# A *fln-2* mutation affects lethal pathology and lifespan in *C. elegans*

Yuan Zhao [1], Hongyuan Wang [1,2], Richard J. Poole [3] & David Gems[1]*

Differences in genetic background in model organisms can have complex effects on phenotypes of interest. We previously reported a difference in hermaphrodite lifespan between two wild-type lines widely used by *C. elegans* researchers (N2 hermaphrodite and male stocks). Here, using pathology-based approaches and genome sequencing, we identify the cause of this difference as a nonsense mutation in the filamin gene *fln-2* in the male stock, which reduces early mortality caused by pharyngeal infection. We show how *fln-2* variation explains previous discrepancies involving effects of *sir-2.1* (sirtuin deacetylase) on ageing, and show that in a *fln-2(+)* background, *sir-2.1* over-expression causes an FUDR (DNA synthesis inhibitor)-dependent reduction in pharyngeal infection and increase in lifespan. In addition we show how *fln-2* variation confounds effects on lifespan of *daf-2* (insulin/IGF-1 signalling), *daf-12* (steroid hormone signalling), and *eat-2* (putative dietary restriction). These findings underscore the importance of identifying and controlling genetic background variation.

---

[1] Institute of Healthy Ageing, Department of Genetics, Evolution and Environment, University College London, London WC1E 6BT, UK. [2] School of Chemistry and Chemical Engineering, Harbin Institute of Technology, Harbin, China. [3] Department of Cell and Developmental Biology, University College London, London WC1E 6BT, UK. *email: david.gems@ucl.ac.uk

Research using invertebrate model organisms has revealed much about how genotype specifies many aspects of phenotype, including ageing. Studies on the short-lived nematode *Caenorhabditis elegans*, in particular, have led to the discovery of signalling pathways that affect lifespan, not only in nematodes but also in higher organisms, including humans[1,2]. These include the insulin/IGF-like signalling (IIS) and the target of rapamycin (TOR) pathways[3,4]. In recent years, the importance of genetic variation in *C. elegans* strain backgrounds has become increasingly clear, partly arising from inadvertent selection acting on the wild type under laboratory conditions and/or random mutation accumulation[5–7]. Such variation may confound the effects of genes of interest.

First isolated more than 60 years ago, the laboratory strain N2 (Bristol) is the standard reference strain for *C. elegans* research[6]. N2 is distributed by the *Caenorhabditis* Genetics Center (CGC) either as a hermaphrodite only stock (which we will refer to here as N2H) or as a male stock (referred to at the CGC as N2 male, which we will refer to as N2M). We previously noted that hermaphrodite lifespan is slightly but consistently longer in N2M than N2H (+11% median lifespan)[8]. As both stocks have been used as wild type, it is possible that the stock difference may have confounded prior studies of gene effects on lifespan.

Ageing limits animal lifespan by promoting multiple pathologies that increase mortality rate. Using necropsy analysis, we recently identified death with a swollen, infected pharynx (P [big P] death) as one form of mortality in *C. elegans*[9]. Under standard laboratory culture conditions[10], initial intra-pharyngeal infection occurs in early adulthood during a period of fast pumping and mastication of the bacterial food source, and later accounts for ~40% of N2H deaths, largely those occurring prior to mean lifespan. The remainder of the population die with an uninfected but often atrophied pharynx (p [small p] death). Changes in the shape of survival curves of different *C. elegans* strains can be understood in terms of altered frequency and/or timing of P and p death (mortality deconvolution)[9].

In this study, we use mortality deconvolution to investigate the cause of the difference in lifespan between N2H and N2M, and the question of which of these N2 lines is wild type. We report that the longer lifespan of N2M is attributable to reduced P frequency that results from a single nonsense mutation in the X-linked gene *fln-2*, indicating that N2H is wild type and N2M mutant. We demonstrate the importance of controlling for *fln-2* genotype by illustrating how it can confound investigations, using several examples of genes that have been the focus of many studies of ageing in *C. elegans*: *sir-2.1*, *daf-2, daf-12* and *eat-2*.

## Results

**N2M longevity is due to reduced P death frequency**. To investigate the causes of the lifespan difference between N2H and N2M, we performed mortality deconvolution on the two lines, comparing timing and frequency of P and p death, using combined mortality and necropsy analysis[9]. Overall mean lifespan of N2M hermaphrodites was 16.6% longer than N2H (22.1 days vs. 18.7 days, $p < 0.0001$, log rank test)(Fig. 1a; Supplementary Table 1), consistent with previous findings[8]. Necropsy analysis revealed that 37% of N2H worms died with P, consistent with previous findings[9], but this was the case in only 13% of N2M worms ($p = 0.0031$, Student's $t$ test)(Fig. 1b). By contrast, the lifespans of both the P and p subpopulations were similar in the two lines (Fig. 1a; Supplementary Table 1). Thus, the lifespan difference between N2H and N2M is attributable to reduced P death frequency in the latter. This predicts that blocking *E. coli* proliferation with antibiotics or UV radiation, which completely

prevents P death[9], should abrogate the difference in lifespan between N2H and N2M, and this proved to be the case (Fig. 1a; Supplementary Fig. 1a; Supplementary Table 1).

To further investigate the difference between N2H and N2M we compared other characteristics of the two lines. No difference in self-fertilised brood size was detected, consistent with previous findings[8] (Supplementary Fig. 1b). Mutants with reduced pumping rate often show reduced P frequency[9], raising the possibility that pumping rate might be reduced in N2M, but it was not (Supplementary Fig. 1c). Moreover, the difference between N2H and N2M is not attributable to effects of parental mating in the latter: hermaphrodite progeny of mated N2H still show higher frequency of P death than N2M (Supplementary Fig. 1d). Thus, N2M's intrinsic resistance to pharyngeal *E. coli* infection accounts for its longer lifespan.

We previously reported evidence that pharyngeal infection occurs in two stages: initial bacterial invasion occurs during early adulthood, apparently due to activity-dependent damage to the pharyngeal cuticle, while major infection occurs later in some animals when bacteria escape cyst-like intracellular structures and proliferate[9] (for examples of different stages of infection, see Supplementary Fig. 2; Supplementary Movie 1). To test whether each stage can be selectively suppressed, worms were fed with *E. coli* OP50 expressing a red fluorescent protein (OP50-RFP), which renders initial bacterial invasion visible as red fluorescent puncta within pharyngeal tissue[9] (Fig. 1c). Culture on antibiotic-treated *E. coli* did not suppress initial bacterial invasion of the pharynx (presumably because antibiotics do not suppress grinder-generated mechanical damage and perforation of the pharyngeal cuticle[9]) but did prevent later bacterial proliferation, while in pumping-defective *phm-2* mutants initial invasion was fully suppressed (Fig. 1c), perhaps due to a combination of reduced mechanical damage and bacterial lawn avoidance[11]. Thus, it is possible to selectively suppress the invasion and proliferation stages.

We then asked: which stage is suppressed in N2M? Initial bacterial invasion was observed in 40% of the population in N2M, compared to 75% in N2H as early as day 4 of adulthood; yet in both strains about half of those showing early signs of invasion die with a swollen, infected pharynx (Fig. 1d). These results suggest that N2M is resistant to pharyngeal cuticle perforation and/or initial bacterial invasion rather than to later bacterial proliferation.

**N2M longevity is caused by *fln-2(ot611)***. Reduced P frequency in N2M likely reflects a different genotype. Analysis of F1 hybrids from a N2H/N2M cross showed that the N2M infrequent P trait is recessive (Fig. 1e). We also noted, during backcrossing with N2M of a strain with high frequency of P, that P frequency dropped in a single round of outcrossing from high to low P frequency (Fig. 1f), suggesting effects of a single gene.

To identify the causative variant, whole genome sequencing was used. An initial comparison between genomic sequences of the two N2 lines revealed 4259 single nucleotide polymorphisms (SNPs) and small insertions and deletions (INDELs). Most of these mutations likely accumulated during separate propagation of N2H and N2M before they were both frozen in 1975[8]. In order to map the mutation affecting P frequency, we performed Variant Discovery Mapping[12]. For this, N2H and N2M were crossed and the progeny of individual F2 recombinant lines with the N2M phenotype pooled and subjected to genome sequencing (Fig. 2a). Parental allele enrichment peaked near the 10 Mb position on LGX implying the presence of the causal variant in this vicinity (Fig. 2b, c; Supplementary Figs. 3 and 4a). Variants within this region co-segregated with low P frequency in the aforementioned

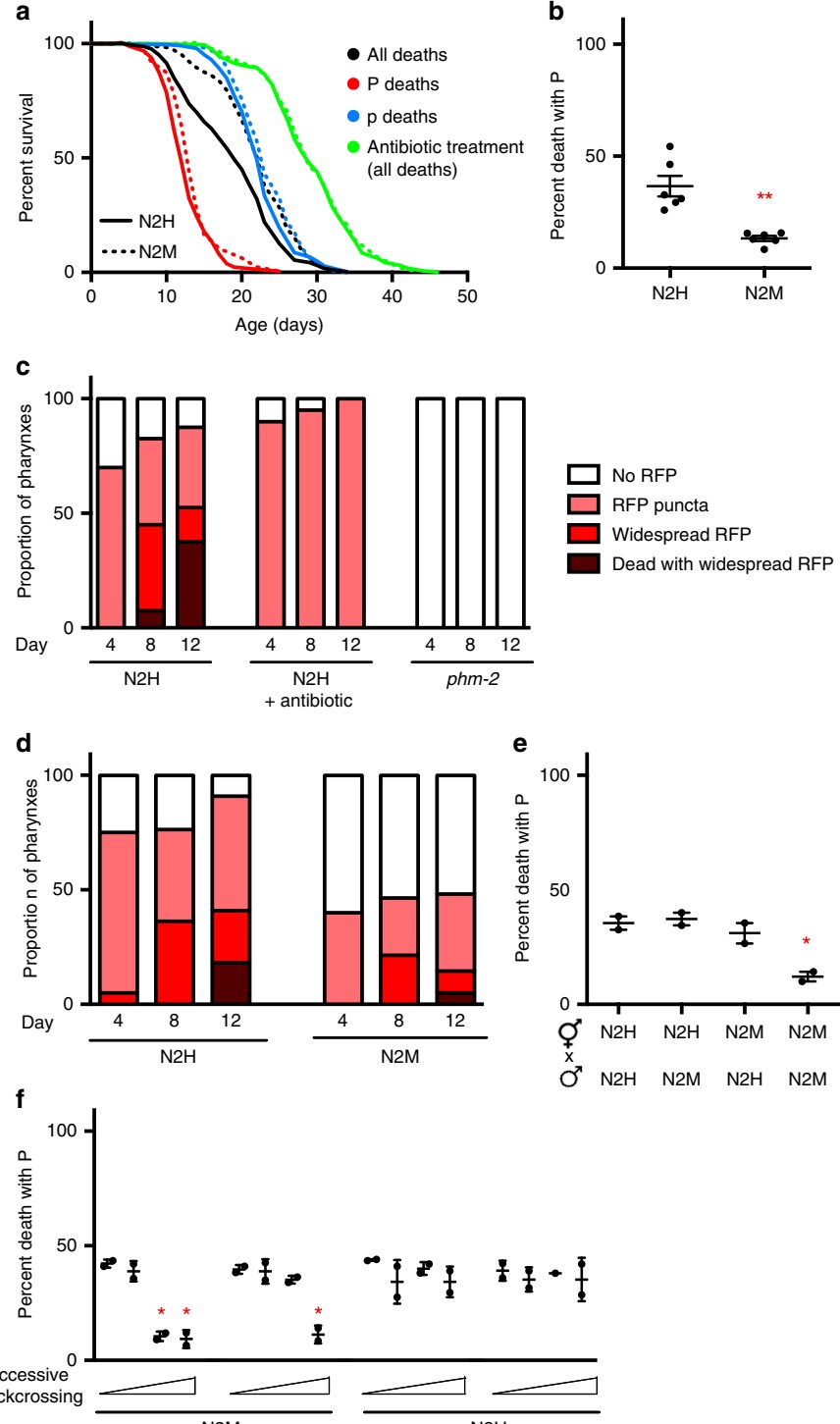

**Fig. 1** N2M is resistant to pharyngeal infection. **a** N2M (dashed lines) is longer lived than N2H (solid lines) (+16.6%, $p < 0.0001$, log rank test), yet neither P or p N2M subpopulations are long lived. N2M longevity disappears after treatment with antibiotics. Whole population in black, P death sub-population in red, p death in blue, and antibiotic-treated in green. See Supplementary Table 1 for full statistics, and Supplementary data 1 for raw mortality data. **b** Necropsy analysis of the two N2 lines. N2M experience less P death than N2H. Error bars represent s.e.m. of six trials. $p = 0.0031$, Student's $t$ test. **c** Antibiotic and *phm-2* mutants suppress P at different stages. Pharyngeal invasion and widespread infection were scored using high magnification epifluorescence microscopy in worms fed RFP-expressing bacteria. No detectable RFP within pharyngeal tissue is shown in white, initial invasion with RFP puncta in pink, widespread RFP in red, and worms that have died with swollen pharynxes in maroon. Mean of two trials, $n = 18$–20 per time point in each trial. **d** N2M is more resistant to initial bacterial invasion. Colour coding as in (**c**). Mean of two trials, $n = 20$–25 animals per time point in each trial. **e** Reduced P frequency in N2M is not a consequence of parental mating. Error bars represent s.d. of two biological replicates. $p = 0.0234$ for N2M × N2M mating progeny, Student's $t$ test. **f** Resistance to P death following successive rounds of backcrossing is consistent with Mendelian segregation of a single mutant locus. Error bars represent s.d. of two biological replicates. Asterisk indicates $p < 0.05$, Student's $t$ test. See Material and Methods for detail. Source data for (**b**–**f**) are provided as a Source Data file

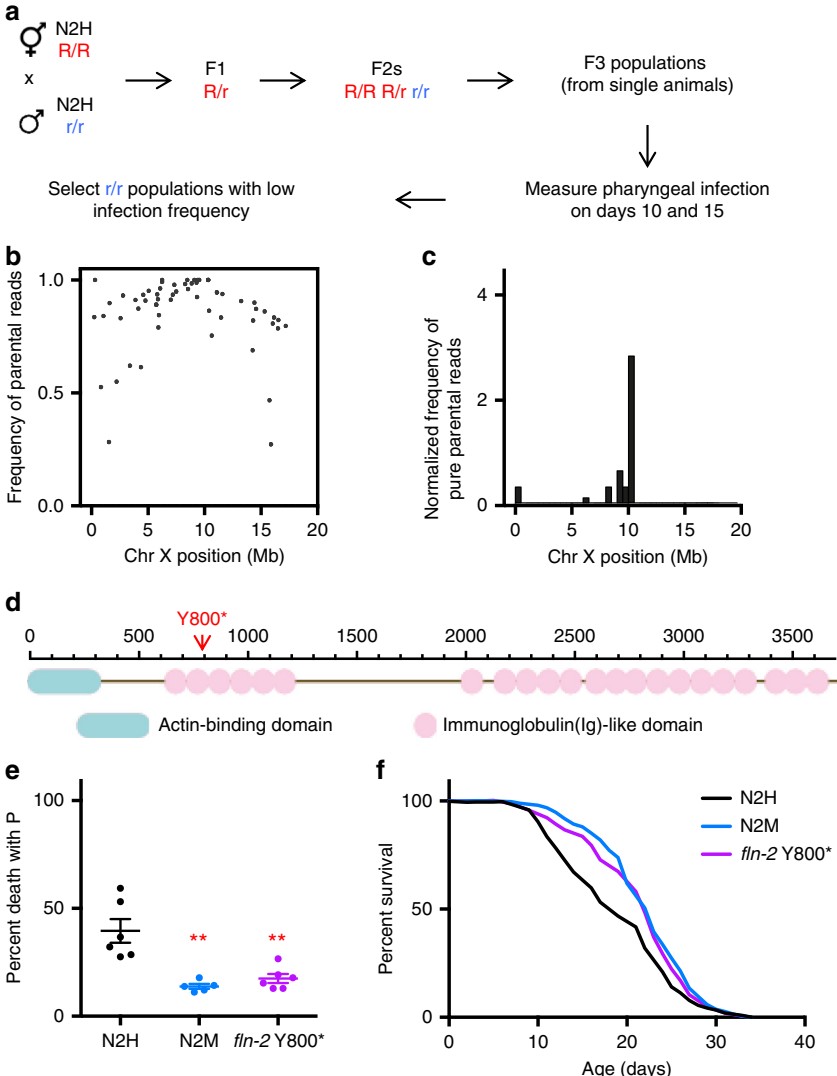

**Fig. 2** Resistance to P death is caused by a recessive mutation in *fln-2*. **a** Design of Variant Discovery Mapping strategy to select the recessive low P phenotype of N2M in F3 progeny. **b** Frequency of parental (N2M genotype) reads on chromosome X. For results for all chromosomes see Supplementary Fig. 3. **c** Normalised frequency of pure parental (N2M genotype) reads on chromosome X. **d** Illustration of the *fln-2(ot611)* mutation on FLN-2A, the longest isoform of *fln-2*. **e** Necropsy analysis and **f** lifespan of N2H-derived *fln-2(syb202)* Y800* strain. N2H survival curve in black, N2M in blue, and *fln-2* Y800* in purple. Asterisks indicate *p* < 0.01, Student's *t* test. Error bars represent s.e.m. of three trials. See Supplementary Table 2 for full statistics. See Supplementary data 1 for raw mortality data. Source data are provided as a Source Data file

backcrossed strains (Fig. 1f), as established by sequencing (16 strains sequenced). The 3 Mb region near the parental allele enrichment peak contained four genetic differences predicted to alter protein sequence, including three missense and one nonsense mutation (Supplementary Fig. 4b). The nonsense mutation was in *fln-2*, which encodes a filamin that is localised, among other places, to the pharyngeal cuticle[13]. We therefore focused on *fln-2*.

FLN-2 is a distant ortholog of filamin-A (FLNA) in humans, which links actin filaments to membranes and is involved in cytoskeletal remodelling and mechanical force sensing. The variant in *fln-2* results in a stop codon at Y800 in N2M, downstream of the N-terminal actin-binding domain in the second immunoglobulin(Ig)-like domain[13] (Fig. 2d). Of the 16 isoforms that *fln-2* encodes, the premature stop codon affects six, all of which contain the actin-binding domain (Supplementary Fig. 4c). We checked the *fln-2* allele in other *C. elegans* wild isolates available at the *Caenorhabditis elegans* Natural Diversity Resource (CeNDR)[14], and found that none of the wild strains

contain the Y800* allele. *fln-2* orthologs in a selection of other nematode species also encode the full-length *fln-2(+)* allele and not the Y800* equivalent (Supplementary Fig. 4d). We therefore conclude that the *fln-2* variant in N2M is a nonsense mutation. This mutation was previously observed 6 out of 17 sequenced EMS-mutagenized genomes[15] and designated *fln-2(ot611)* (formerly *flnb-1*); on the assumption that strains containing the mutant allele were derived from N2M, we will refer to the variant identified here as *fln-2(ot611)*.

To test whether *fln-2(ot611)* causes the N2M infrequent P trait, we first tested effects of the mutation *fln-2(ok1305)*, which contains a deletion in the N-terminal region. Although it showed infrequent P death (Supplementary Fig. 5a), the strain RB1240 also contained *ot611*, making this result difficult to interpret. Another *fln-2* mutant allele, *fln-2(tm4687)*, which does not contain *ot611*, has a deletion in the C-terminal region affecting all *fln-2* isoforms. This also caused low P frequency (Supplementary Fig. 5b), demonstrating that variation at *fln-2* can affect P incidence. Additionally, we introduced the Y800* mutation into

N2H using CRISPR/Cas9. The resulting *fln-2(syb202)* Y800\* allele led to recapitulation of the N2M phenotype, reducing P frequency from 45 to 19%, with little effect on p lifespan (Fig. 2e, f; Supplementary Table 2). Taken together, these results indicate that *fln-2(ot611)* is the cause of the reduced P frequency of N2M. Tests on four *C. elegans* wild isolates showed a P death incidence similar to that of N2H (Supplementary Fig. 5c), suggesting that this relatively high P frequency is a species-typical (i.e. wild-type) characteristic of *C. elegans*.

Although N2H is predominantly used as the wild-type strain in *C. elegans* research (requested from the CGC 3,633 times compared to 173 times for N2M [2006–2016]), N2M is sometimes used for strain construction or backcrossing. As a limited test of the presence of *fln-2(ot611)* among *C. elegans* strains, we genotyped the *fln-2* allele in a selection of strains present in our strain collection. This revealed the presence of *fln-2(ot611)* in 23/50 of cases, including some strains generated by the *C. elegans* Gene Knockout Project (*gk* and *ok* alleles)(*ot611* present: VC prefix strains, 6/10; RB prefix strains, 6/6) and the *C. elegans* Expression Project (*s* alleles)(*ot611* present: 9/9)[16,17] (Supplementary Table 3).

**Discrepant effect of *sir-2.1* attributable to *fln-2* variation.** Given the effect of the *fln-2(ot611)* allele on pharyngeal pathology and lifespan, and the fact that the Gems lab routinely used N2M as the wild-type for backcrossing[8], we set out to revisit some earlier, puzzling results. In budding yeast (*Saccharomyces cerevisiae*) over-expression of the NAD-dependent histone deacetylase *SIR2* (sirtuin) increases replicative lifespan[18]. There have been conflicting reports about whether or not increasing sirtuin levels in *C. elegans* increases lifespan. In *C. elegans*, independent transgenic lines with *sir-2.1* over-expression (oe) generated in two studies were long lived[19,20]. In the case of one high copy number *sir-2.1* over-expresser (transgene array *geIs3*, hereafter referred to as *sir-2.1(oe)*)[19] it was subsequently reported that longevity of the original strain (LG100) was attributable to a second site *dyf* gene mutation, and that *sir-2.1(oe)* no longer increased lifespan after backcrossing with N2M[21]. However, in another study that confirmed the presence and action of the second site mutation, a residual, smaller lifespan-increasing effect of *sir-2.1(oe)* was seen after outcrossing with N2 (line not specified)[22].

We asked: could this discrepancy be attributable to effects of N2H vs. N2M background differences? Genotyping of the strains used by Viswanathan and Guarente (2011)[22] and Burnett et al. [21] confirmed that the former were *fln-2(+)*, and the latter *fln-2(ot611)*, presumably reflecting the different N2 lines used in backcrossing of the original *sir-2.1(oe)* line. Consistent with both previous reports, *sir-2.1(oe)* resulted in increased lifespan in a *fln-2(+)* background (+27% mean lifespan, *p* < 0.0001, log rank test), but not a *fln-2(ot611)* background (*p* = 0.96, log rank test) (Fig. 3a; Supplementary Fig. 6a; Supplementary Table 4) under culture conditions as previously described, in which the thymidylate synthase inhibitor 5-fluorodeoxyuridine (FUDR) is added to block progeny development (see Methods and below).

Mortality deconvolution revealed that there was no significant difference in the frequency of P death in a *fln-2(ot611)* background (GA strains: GA468 *geIs3* [*sir-2.1(+) rol-6*]; *fln-2 (ot611)* compared to GA707 *wuEx166* [*rol-6*]; *fln-2(ot611)*)), but that *sir-2.1(oe)* significantly reduced P death frequency in a *fln-2 (+)* background (LG strains: LG394 *geIs3* [*sir-2.1(+) rol-6*] compared to LG398 *geIs101* [*rol-6*]) (Fig. 3b; Supplementary Fig. 6b). All four strains were then backcrossed with N2H, which for the GA strains replaced *fln-2(ot611)* with *fln-2(+)*, after which *sir-2.1(oe)* reduced P death frequency and increased lifespan in both sets of strains (+19% and +32% mean lifespan, respectively)

(Fig. 3c, d; Supplementary Table 5). In a *fln-2(+)* background, the increase in lifespan resulting from *sir-2.1(oe)* was the result not only of reduced P death frequency, but also of a small but statistically significant increase in p lifespan (+14%, +13% mean lifespan in GA strains and LG strains, *p* = 0.0016 and 0.0002, respectively, log rank test) (Supplementary Fig. 6c; Supplementary Table 5).

In the previous studies, FUDR was used to block progeny development: 1.6 mM in the 2001 study[19], and 10 to 50 μM[21] or an unspecified concentration[22] in the 2011 studies. Therefore in the trails described above FUDR was included (10 μM in Fig. 3a–d, and 50 μM in Supplementary Fig. 6a, b). Since impacts of many interventions on lifespan are affected by the presence of FUDR[23–26], and since fluoropyrimidines have complex effects both on *E. coli* and *C. elegans*[27], we also tested effects of *sir-2.1(oe)* on P frequency and lifespan (overall and P or p alone) in the absence of FUDR. Here no effect was detected in either *fln-2* background, in either the original strains or the outcrossed strains (2 trials each) (Fig. 3e, f; Supplementary Fig. 6d, e; Supplementary Tables 6 and 7).

We further tested for interaction between FUDR and *sir-2.1* over-expression using a previously described low copy number transgene array *pkIs1642*[20]. It was previously reported that longevity resulting from *pkIs1642 sir-2.1(oe)* was abrogated by outcrossing[21]. Here both original and outcrossed strains were *fln-2(+)*. Nonetheless, we observed that the longevity of the non-outcrossed *pkIs1642* strain was, again, FUDR dependent (Supplementary Fig. 7; Supplementary Table 8). However, here longevity resulted from increased p lifespan rather than reduced P frequency. For more details, see Supplementary Fig. 6.

We also asked whether loss of *sir-2.1* increases P frequency, using the *sir-2.1(ok434)* null allele that harbours a deletion that disrupts the sirtuin catalytic domain[28,29]. After backcrossing into a *fln-2(+)* background, *sir-2.1(ok434)* reduced overall lifespan independently of FUDR (−8% and −9% mean lifespan, *p* = 0.0043 and 0.0425, with and without FUDR respectively, log rank test), consistent with previous reports[20,28]. However, the manner in which *sir-2.1(ok434)* shortened lifespan was FUDR dependent: with FUDR it increased P death frequency, but without FUDR it reduced p lifespan (Supplementary Fig. 8a, b; Supplementary Table 9). Thus, again, the impact of *sir-2.1* on worm ageing is FUDR dependent.

The FUDR dependence of the reduced P death caused by *sir-2.1 (oe)* suggests that SIR-2.1 and FUDR act synergistically to protect against bacterial infection of the pharynx. To explore the mechanism of this protection we compared the progression of pharyngeal infection in OP50-RFP-fed worms with either *sir-2.1 (oe)*, FUDR, or both. Without FUDR, as expected, *sir-2.1(oe)* had little effect on either the initiation or subsequent spread of *E. coli* infection; with FUDR, the appearance of intracellular RFP puncta was unaffected by *sir-2.1(oe)*, but the progression of infections in older animals was suppressed, with fewer animals showing widespread RFP (Fig. 3g). Notably, FUDR alone delayed the appearance of RFP puncta, though it had little effect on the frequency of P (Fig. 3g). Thus, FUDR and *sir-2.1(oe)* affect pharyngeal infection by different mechanisms: FUDR delays initial invasion, while SIR-2.1 suppresses later spread of infection. This suggests that FUDR-induced resistance to bacterial pathogenicity enables SIR-2.1 to reduce later *E. coli* proliferation, leading to synergy between the two interventions. Consistent with this, pre-treatment of *C. elegans* with FUDR increases resistance to the Gram-negative bacterial pathogen *Pseudomonas aeruginosa*[30].

**fln-2 variation confounds effects of *daf-12* on lifespan.** The *daf-12* gene encodes a steroid hormone receptor, and has been shown

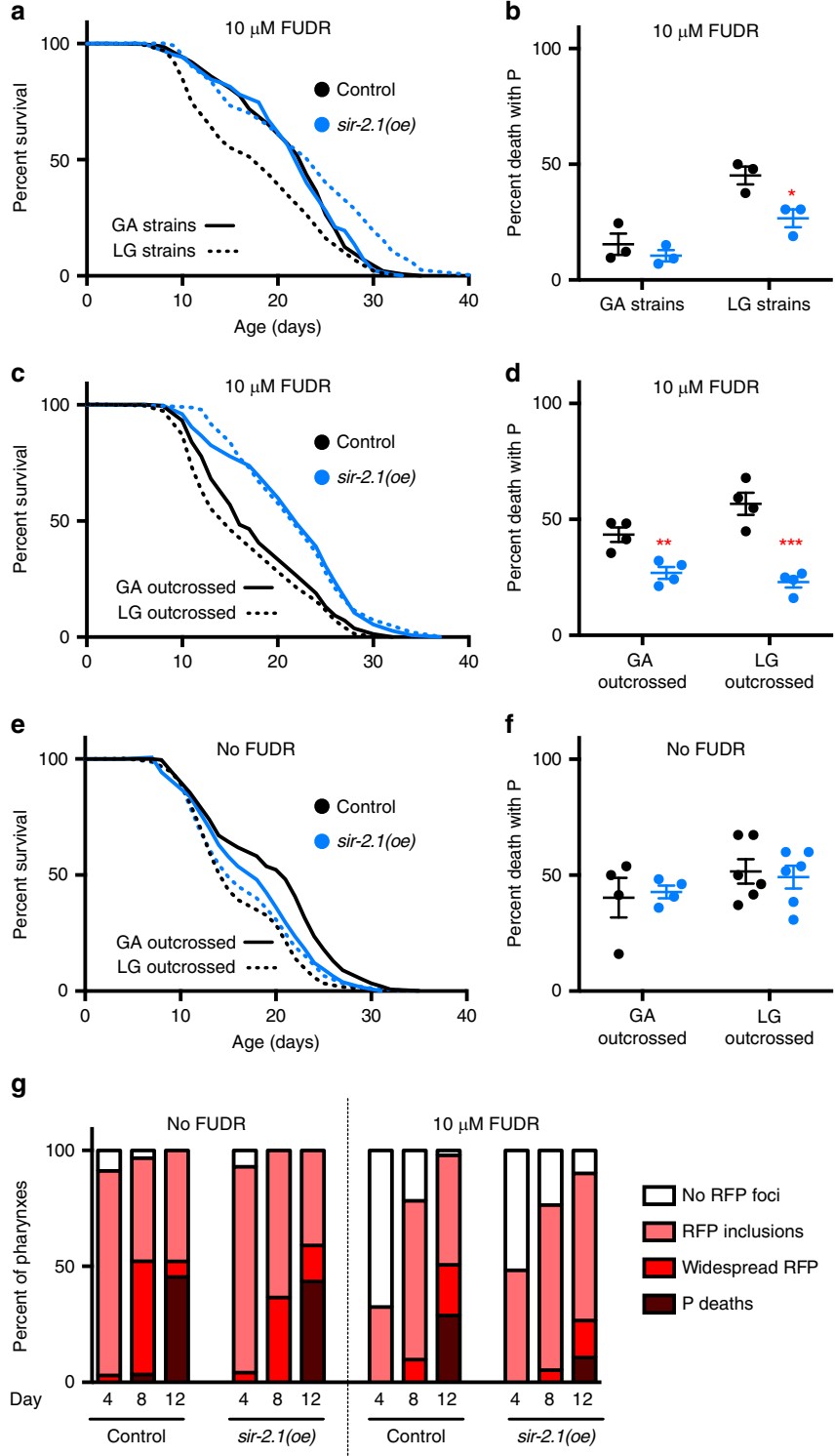

to have complex effects on different *daf-2* insulin/IGF-1 receptor mutants[31–33]. For example, *daf-2(m41)* longevity (the Age phenotype) is partially suppressed by *daf-12(m20)* under certain conditions[31]. We noted that *daf-12(m20)* shortens *daf-2(m41)* lifespan by increasing early mortality[31], and that the *daf-12* and *fln-2* loci are closely linked on LGX (at X:2.36 ± 0.007 cM and X:1.18 ± 0.015 cM, respectively). This raised the possibility that some observed effects of *daf-12* on *daf-2* Age might be attributable to the differences in P frequency due to variation at the *fln-2* locus.

We genotyped *daf-2(m41)* and *daf-2(m41); daf-12(m20)* double mutant strains used in earlier studies[31,32], and found that the *daf-2(m41)* strain, but not the *daf-2; daf-12* strain, contained *fln-2 (ot611)*. Consistent with previous findings[31], *daf-2; daf-12* lifespan was shorter than *daf-2; fln-2*; this was due mainly to a 9-fold higher P frequency in *daf-2; daf-12* (Fig. 4a, b). Comparison of *daf-2; daf-12* and *daf-2* after removal of *fln-2 (ot611)* by outcrossing showed a much smaller but statistically significant 60% increase in P frequency (Fig. 4b). Notably, *daf-12* had little effect on *daf-2(m41)* p lifespan (Fig. 4c); moreover, *fln-2*

**Fig. 3** Reduction of P death by *fln-2(ot611)* masks effects of high copy number *sir-2.1* over-expression *(oe)* on lifespan. **a** Lifespan and **b** necropsy analysis of *sir-2.1(oe)* and control strains used in previous studies. GA strains are indicated by solid lines: GA468 *gels3 sir-2.1(oe)*; *fln-2(ot611)* (blue) and GA707 *wuEx166; fln-2(ot611)* control (black). LG strains are indicated by dashed lines: LG394 *gels3 sir-2.1(oe)* (blue) and LG398 *gels101* control (black). *sir-2.1(oe)* reduced P death and extended lifespan in LG strains but not GA strains, consistent with both previous, conflicting studies[21,22]. Error bar represents s.e.m. of three trials. See Supplementary Table 4 for full statistics. See Supplementary data 1 for raw mortality data. Source data are provided as a Source Data file. **c** Lifespan and **d** necropsy analysis of *sir-2.1(oe)* and control strains after backcrossing four times with N2H. GA outcrossed in solid lines: GA1909 *gels3 sir-2.1(oe)* (blue) and GA1907 *wuEx166* control (black); LG outcrossed in dashed lines: GA1913 *gels3 sir-2.1(oe)* (blue) and GA1915 *gels101* control (black). *sir-2.1 (oe)* strains (blue) from both labs showed a reduced frequency of P death compared to control strains (black) in *fln-2(+)* background. See Supplementary Table 5 for full statistics. **a–d** Trials conducted in the presence of 10 μM FUDR. See Supplementary data 1 for raw mortality data. **e** Lifespan and **f** necropsy analysis of backcrossed *sir-2.1(oe)* and control strains, with FUDR excluded. Colour coding as in (**c**, **d**). See Supplementary Table 7 for full statistics. See Supplementary data 1 for raw mortality data. **g** *sir-2.1(oe)* and 10 μM FUDR interact to protect against progression of bacteria infection. Pharyngeal invasion and widespread infection were scored in worms fed RFP-expressing bacteria. Colour coding as in Fig. 1c. Mean of two trials, n = 16–20 per time point in each trial

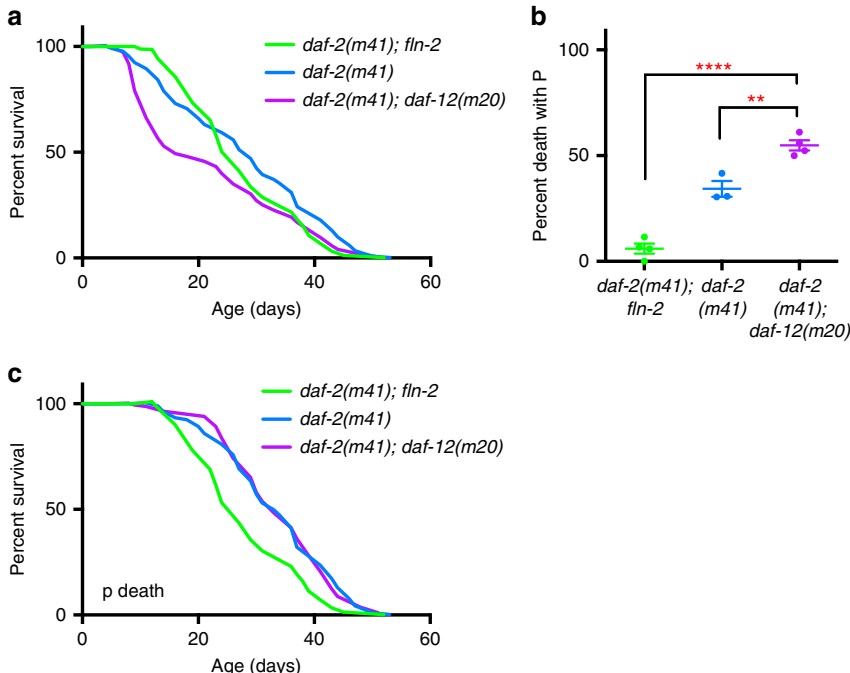

**Fig. 4** Effect of *daf-12* on *daf-2* at 25 °C is partially affected by *fln-2(ot611)* variation. **a** Lifespan and **b** necropsy analysis of *daf-2(m41); daf-12(m20)* and *daf-2 (m41)* mutants in *fln-2(+)* or *fln-2(ot611)* backgrounds. *daf-2 fln-2* in green, *daf-2* in blue, and *daf-2 daf-12* in purple. Asterisks indicate significant differences, *p* < 0.0001 and 0.01, respectively. Error bars represent s.d. of two trials. See Supplementary Table 10 for full statistics, and Supplementary data 1 for raw mortality data. Source data are provided as a Source Data file. **c** Lack of lifespan effect by *daf-12* in *daf-2* p sub-population lifespan in *fln-2(+)* background. See Supplementary data 1 for raw mortality data

*(ot611)* significantly reduced *daf-2(m41)* p lifespan (Fig. 4c; Supplementary Table 10). We therefore conclude that while *daf-12* does partially suppress the *daf-2(m41)* Age, it acts solely by increasing P frequency, an effect that appeared greater due to the presence of *fln-2(ot611)* in the *daf-2* mutant. The fact that the lifespans of *daf-2* and *daf-2; fln-2* nematodes were similar was due to antagonistic effects of *fln-2* on P frequency (increasing lifespan) and p survival (reducing lifespan) in this condition. These results further illustrate how *fln-2* variation can confound studies of lifespan genetics.

**fln-2(ot611) masks eat-2 Age effect.** Dietary restriction (DR) increases healthspan and lifespan in many model organisms[34]. In *C. elegans*, some Eat mutants with reduced feeding rate (e.g. *eat-2*) show increased lifespan, which has been interpreted as resulting from DR[35]. In previous work in this lab with *eat-2* alleles crossed into an N2M background, we often saw little effect of *eat-2* on lifespan (A. Benedetto, M. Keaney, J.M.A. Tullet, and

D. Gems, unpublished). More recently we showed that many Eat mutants show reduced P frequency, which may reflect reduced mechanical damage to the pharynx otherwise caused by high pumping rate in early life[9]. Mortality deconvolution of *eat-2* populations showed that the lifespan increase is largely attributable to reduced P death frequency. This predicts that *eat-2* should increase lifespan more in a N2H background than a N2M background, due to the higher P frequency in the former. To test this, we performed mortality deconvolution on *eat-2(ad1116)* mutants in *fln-2(+)* and *fln-2(ot611)* backgrounds. As expected, P death frequency was reduced by either *fln-2* or *eat-2* (Fig. 5a). Notably, *eat-2* increased lifespan more in a *fln-2(+)* background than in a *fln-2(ot611)* background (mean lifespan +27% and +9%, *p* < 0.0001, *p* = 0.0043, respectively, log rank test) (Fig. 5b; Supplementary Table 11), while the effect on p lifespan was modest in both backgrounds (+8% and +5%, *p* = 0.0183 and 0.0149, respectively, log rank test) (Fig. 5c). When P was eliminated using antibiotics, effects of *fln-2* genetic background disappeared, as expected (Fig. 5d).

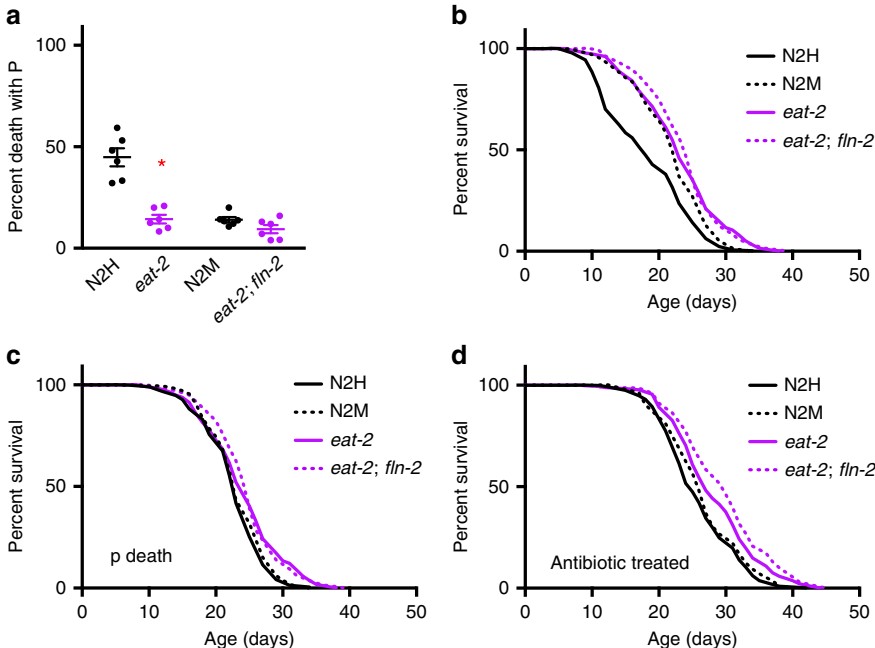

**Fig. 5** Reduction of P death by *fln-2(ot611)* masks lifespan effects of *eat-2*. **a** Necropsy analysis and **b** lifespan of *eat-2(ad1116)* mutants in *fln-2(+)* or *fln-2 (ot611)* backgrounds. N2H survival curve indicated by solid black line, N2M by dashed black line, *eat-2* by solid purple line, and *eat-2 fln-2* by dashed purple line. Error bars represent s.e.m. of three trials. See Supplementary Table 11 for full statistics, and Supplementary data 1 for raw mortality data. Source data are provided as a Source Data file. **c** *eat-2(ad1116)* causes a small but significant increase in p sub-population lifespan in both *fln-2(+)* and *fln-2(ot611)* backgrounds. See Supplementary data 1 for raw mortality data. **d** Effects of *fln-2(ot611)* on lifespan in *eat-2(ad1116)* mutants disappears after treatment with antibiotics (200 μg/ml carbenicillin). See Supplementary data 1 for raw mortality data

## Discussion

The findings reported here reveal a major confounding variable in studies of ageing in *C. elegans*: variation at the *fln-2* locus between two widely used N2 lines. We demonstrate how this has actually confounded research on *C. elegans* ageing, distorting experimental results, and causing confusion and controversy, issues that can now be readily resolved. These results also provide further evidence of the power of mortality deconvolution and, more broadly, pathology-centred approaches to solve complex problems in biogerontology.

The advance of science often requires a process of over-coming limitations within a field, both conceptual and technical. The latter includes identification and control of confounding variables, which, over time, can lead to accumulation of unsolved questions and conflicting results. This is a particular problem with model organism biogerontology, since lifespan as a trait is particularly sensitive to uncontrolled variables[36,37]. For *C. elegans*, these include variables affecting genetic background, such as that between N2H and N2M, and effects of variation at *fln-2* as described here. It also includes a range of environmental variables; since the introduction of a standardised *C. elegans* plate culture system[10], several modifications have been introduced, including the use of FUDR to block progeny production, and antibiotics or UV irradiation to block *E. coli* proliferation. It is becoming increasingly clear that the effects of many interventions that alter *C. elegans* lifespan reflect, at least partially, changed interactions between *C. elegans* and *E. coli*[9,38,39]. Thus, variables affecting *C. elegans*-*E. coli* interactions, including *fln-2* allele variation and use of different culture conditions, either introduce confounding variables, or make it difficult to relate results from different research groups to one another.

Variation at *fln-2* can confound studies of the genetics of lifespan, as illustrated here using three cases, involving *sir-2.1*, *daf-2* and *daf-12*, and *eat-2*. It is therefore possible that variation at *fln-2* has also confounded other studies of *C. elegans* ageing.

Our results resolve a controversy surrounding effects of *sir-2.1* over-expression in *C. elegans* lifespan[21,22,40]. This provides a useful lesson in how science can be impeded by confounding variables, in this case three of them. The first was the presence of a *dyf* mutation which increased lifespan in a *sir-2.1(oe)* strain[21,22]. The second was the presence of *fln-2(ot611)* in strains back-crossed with N2M, which suppressed the effect of *sir-2.1(oe)* on lifespan[21]. The third was the use of FUDR, which as we show here is required for *sir-2.1(oe)* to increase lifespan; it is notable that FUDR-dependent induction of increased lifespan by hypertonic stress is partially sirtuin dependent[24]. Does this FUDR-dependence invalidate the conclusion that *sir-2.1(oe)* protects against ageing? This question raises important issues about the use of *C. elegans* as a model for understanding human ageing. We have argued elsewhere[41,42] for a reconceptualization of the relationship between ageing and lifespan, as follows. Lifespan is not a read out of one process, i.e. ageing, but rather a function of some (but not all) senescent pathologies that arise as animals age. Importantly, the identity of the pathologies that limit lifespan will vary not only between species, but within species depending upon genotype and environment[41,42]. According to this view, the key objective for understanding ageing is to understand the etiologies of any pathology, whether they affect lifespan or not. In terms of *sir-2.1(oe)*, FUDR and lifespan, if the use of FUDR enables studies that yield useful insights about senescent pathology, then this is a good model to study.

We showed here that under standard culture conditions (proliferating *E. coli*, no FUDR), previously reported suppression of *daf-2(m41)* Age (25 °C) by *daf-12(m20)* is likely to have been partially attributable to variation at *fln-2* (Fig. 4a–c). There is other evidence that signalling via the DAF-9/DAF-12 pathway can promote *daf-2* Age. For example, at 22.5 °C *daf-9(rh50)* shortens *daf-2(e1368)* lifespan[43], and *daf-2(e1368)* Age can be enhanced by treatment with exogenous DAF-12 ligand (Δ⁴-dafachronic acid)[44]. However, our results imply that clearer

understanding of how the IIS and DAF-12 steroid signalling pathways interact to affect ageing could be achieved by enhanced control of confounding variables, including *fln-2* and culture condition variables (as listed above).

These results reveal that the greater lifespan of N2M is attributable to *fln-2(ot611)*. The presence of the *fln-2(+)* allele in N2H and all *C. elegans* wild isolates examined demonstrates that N2H is the wild type and N2M a long-lived mutant, in contrast to a previous suggestion[8]. This is an argument for discontinuing the use of N2M as wild type for most purposes. The mechanisms by which *fln-2(ot611)* protects against pharyngeal infection remain to be explored. Comparison between the two N2 strains raised on fluorescent *E. coli* suggests that *fln-2(ot611)* is resistant to cuticle perforation and/or bacteria invasion. The presence of FLN-2 protein in the pharyngeal cuticle[13] is consistent with a role as a direct, cell autonomous determinant of susceptibility to mechanical damage and/or bacterial invasion.

## Methods

**Culture methods and strains.** *C. elegans* maintenance was performed using standard protocols[10]. Strains were grown at 20 ˚C on NGM agar plates seeded with *E. coli* OP50. *C. elegans* strains used: N2 and N2 male stocks recently obtained from the CGC (designated N2H and N2M, respectively), CF1038 *daf-16(mu86)*, RB1240 *fln-2(ok1305ot611)*, FX4687 *fln-2(tm4687)*, UN1066 *xbEx0816 [fln-2c::gfp]*, GA1947 *fln-2(syb202)*, GA468 *fln-2(ot611) geIs3 [sir-2.1(+)+rol-6(su1006)]*, GA707 *fln-2(ot611) wuEx166 [rol-6(su1006)]*, LG394 *geIs3 [sir-2.1(+)+rol-6 (su1006)]*, LG398 *geIs101 [rol-6(su1006)]*, NL3908 *unc-119(ed3) pkIs1641(unc-119 [+])*, NL3909 *unc-119(ed3) pkIs1642(unc-119[+] sir-2.1[+])*, GA1934 *sir-2.1 (ok434)*, GA1928 *daf-2(e1370)*, DR1296 *daf-2(e1370) daf-12(m20)*, GA1945 *daf-2 (m41)*, DR1547 *daf-2(m41) daf-12(m20)*, DA1116 *eat-2(ad1116)*, GA66 *eat-2 (ad1116) fln-2(ot611)*. Strains genotyped with respect to *fln-2* allele are listed in Supplementary Table 3.

**Lifespan assays and necropsy analysis.** Unless otherwise specified, lifespan assays were conducted at 20 ˚C and without FUDR, and animals were transferred daily during the egg laying period, and every 3-7 days thereafter. Animals lost due to causes other than senescence (e.g. desiccation on the side of the plate, internal hatching, extensive rupture through the vulva) were censored. For lifespans performed at 25 ℃ (*daf-2 daf-12* experiments), worms were raised at 20 ℃, and transferred to 25 ℃ at L4 stage. For trials performed in the presence of FUDR, L4 stage animals were transferred to seeded NGM plates containing the specified concentration of FUDR, and animals were transferred every 3–7 days to avoid hypertonic stress[24]. For antibiotic treatment, 200 μg/ml carbenicillin was added to two-day old lawns of OP50, and nematodes were transferred to antibiotic plates as L4 larvae. Mortality and pharyngeal status of corpses were scored every 2–3 days. For lifespan analysis on UV-irradiated *E. coli* OP50, bacteria were inoculated onto NGM plates and allowed to grow overnight at room temperature before exposure to UV light (Stratagene UV Stratalinker 2400) for 5 min, 2 days prior to use (to detect and exclude plates with bacterial growth). Bacteria-free (alkaline hypochlorite-treated) *C. elegans* eggs were then plated on irradiated *E. coli* lawns. Lifespan assays were repeated at least twice, and raw mortality data are provided in the Ziehm tables (Supplementary Data 1). Survival plots were generated in GraphPad Prism using combined lifespan data from multiple trials, with L4 as day 0 on the time scale.

**OP50-RFP infection analysis.** An overnight culture of OP50-RFP in LB supplemented with tetracycline was washed three times in M9 to remove tetracycline, and seeded onto NGM plates two days before use. L4 worms cultured on OP50 bacteria were transferred to OP50-RFP plates, and transferred daily during the reproductive period. Before imaging, live worms were mounted on 2% agar pads and anesthetised using 0.2% levamisole. Fluorescence within pharyngeal tissue were scored at 630x magnification using a Zeiss AxioImager Z2 microscope with a rhodamine filter. A different cohort of worms was imaged for each time point so as to avoid recovering worms from levamisole.

**Test cross to confirm single-locus association.** Strain GR1307 *daf-16(mgDf50)* was backcrossed with either N2H or N2M males, and F1s allowed to self. Single F2 animals were picked at random and the *daf-16* allele genotyped, and F3 populations of single *daf-16* homozygous animals were scored in necropsy analysis and used for further backcrossing. The same analyses were performed for a single F4, F6, and F8 animal of each F2 line. All progeny of the backcrossed F2, F4, F6, and F8 animals used in lifespan were frozen and subsequently genotyped with respect to the *fln-2* allele.

**Whole genome sequencing.** Genomic DNA was extracted using Gentra Puregene Tissue Kit (Qiagen), and sequenced with an average coverage of 100× (Center for Applied Genomics, The Hospital for Sick Children, Toronto, Canada). N2M genome sequence is available on ENA with the accession number PRJEB34655 [https://www.ebi.ac.uk/ena/browser/view/PRJEB34655], and will be incorporated in the N2 genome sequence on CeNDR[14]. For variant discovery mapping, N2H hermaphrodites and N2M males were allowed to mate overnight, after which males were removed. F1 hermaphrodites were allowed to self-fertilise, and a total of 250 F2 animals were singled and allowed to self-fertilise. From each plate, 50–60 F3 progeny were transferred to plates containing 10 μM FUDR, allowed to age, and then scored for pharyngeal phenotype. 50 of the populations displaying lowest frequency of P death were re-analysed and 25 populations that showed less than 20% P death in both tests were selected. F3 and F4 genomic DNA from selected populations were then extracted and pooled to be sequenced. The sequencing data were processed in CloudMap using the variant discover mapping method, which mapped the causal variant to the vicinity of the 10 Mb position on chromosome X.

**Genotyping *fln-2* alleles.** The following primers were used to amplify the genomic region flanking the *fln-2* mutation site: forward 5′-GGTGTTCGATTCTGGTCT GG-3′; reverse 5′-ACATCGACGAGAAGACAACAC-3′. PCR product was sequenced using primer 5′-TGTACCCAGAAATTGACAAGATAC-3′. Allele-specific PCR can be used to discriminate between the genotypes, using the following primers: wild-type-specific forward 5′-taccattccgagcttattgattgttacctGGACG GCGCTGGTCCATAC-3′; *ot611*-specific forward 5′-GGACGGCGCTGGTCTAT AA-3′; ASP reverse 5′- ATCGCATGAACCATAAATGATG-3′; the wild-type PCR product is 30nt longer than the mutant product.

***fln-2(syb202)* Y800* allele.** The allele harbouring the Y800* mutation was generated using co-CRISPR method in N2H (Sunybiotech). sgRNAs TCATT-CACTCCGGACGGCGCTGG and CCACGTTCTTTTTAACAGAATGG were used, and synonymous mutations were made in the sgRNA/PAM site. The final mutated sequence is tcattcactccggacggAgctggtcaatAAaaaatAcaTgttctttttaacagaatgg.

**Statistics.** Survival analyses of whole populations (i.e. P and p deaths) were conducted using GraphPad Prism, including right censoring of animals lost due to causes other than senescence. To test for statistically significant differences between survival curves, the non-parametric log rank test was used. In the case of P or p subpopulations, due to the absence of information about corpse type, deaths due to causes other than senescence were not censored but rather excluded altogether from statistical analysis. Sample sizes and statistics for each experiment are provided in the lifespan tables in the Supplementary file. An unpaired Student's *t* test was used to compare frequencies of P death between different strains and/or conditions.

**Reporting summary.** Further information on research design is available in the Nature Research Reporting Summary linked to this article.

## Data availability

All statistics relating to survival analysis are provided in the Supplementary tables. Raw mortality data are provided in Ziehm tables[9] in Supplementary Data 1. Source data relating to mortality analysis are provided as a Source Data file. N2M genome sequence is available on ENA with the accession number PRJEB34655.

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

## Acknowledgements

The authors thank the *Caenorhabditis* Genetics Center, which is funded by NIH Office of Research Infrastructure Programs (P40 OD010440), for strains used in this study, and A. Daul at the CGC for information. We thank the Mitani Lab National BioResource Project for strains. The authors thank E. Cram, H. Richly and B. Tursun for strains and reagents. This work was supported by a Wellcome Trust Strategic Award to D. Gems (098565/Z/12/Z) and a Wellcome Trust Research Career Development Fellowship to R.J. Poole (095722/Z/11/Z). The authors would also like to thank A. Benedetto for critically reading the manuscript.

## Author contributions

D.G. and Y.Z. conceived the study. H.W. and Y.Z. performed the experiments, analysed and interpreted the data. R.P. performed the Variant Discovery Mapping. D.G. and Y.Z. wrote the manuscript.

## Competing interests

The authors declare no competing interests.
