## [Peer Review File · Nature Communications]

Reviewers' comments:

Reviewer #1 (Remarks to the Author):

In their manuscript titled "A common fln-2 mutation affects lethal pathology and lifespan in *C. elegans*", Zhao and colleagues uncovered a confounding factor that affects lab-to-lab and paper-to-paper variability in *C. elegans* lifespan. The authors took a genetic and rudimentary pathological approach to understand differences in lifespan and mortality in different genetic backgrounds in *C. elegans*. They found that a previously undocumented mutation (fln-2) in a male stock of *C. elegans* (N2M) was responsible for the lifespan differences between two *C. elegans* stocks (males (N2M) and hermaphrodites (N2H)). Perhaps more importantly, the report addresses conflicting reports on the effects of different genes on lifespan, with sir-2.1 overexpression being the most potent example. Specifically, the authors addressed an old controversy on the effect of sir-2.1 overexpression on *C. elegans* lifespan. I would like to thank Dr. David Gems for the honesty and carefulness with which he revisited his Burnett et al, 2011 publication to further clarify the old controversy regarding sir-2.1.

The work is well executed and well communicated. It is mostly technically solid; it is supported by a significant body of experimental evidence. There are other technical concerns regarding the conclusions drawn from antibiotics and a roller background (see major comments below). Overall, it is a nice example of a well written, conscientious study (mostly). The results are not exciting or unexpected, but do address previous disparities between reports caused by flawed experimental design. The prior report's flaw stemmed from the Gems lab practice of outcrossing their animals with the N2M stock instead of just generating males from the N2H, like most *C. elegans* labs of which I am aware. Thus, they compared animals crossed with genetically distinct stocks, with clearly different phenotypes (in retrospect), which ended up having different mutations, causing them to dispute the effects of sirtuin overexpression. Given that the stocks were supposed to be isogenic, this seems forgivable, but also seems like it could have been avoided by more cautious experimentation (deriving males from the strains they have). The importance of genetic background has been abundantly clear for years in *Drosophila*, and to all geneticists. Again, in my experience, most people do not use the *C. elegans* male stock, they just generate males from their existing N2H stock. So, the broad relevance of differences in these stocks, even to gerontologists, is questionable. Yet, Nat Comm is for specialists and the relevance of clarifying conflicting Nature reports on the effects sirtuin overexpression in this system, especially from the investigator who disputed it before, may bring the relevance of the manuscript to a higher level. I think it might be more suitable for Scientific Reports, because we are not really learning any new exciting biology, but my peers and the editor might disagree. And I am a bit on the fence about the publication at Nat Comm, especially given the 2017 publication on the same subject in worms (Lucanic et al); yet, it might be important to show this as an example of science self-correcting. Regardless of where it ends up, there are some technical concerns to address about the biological conclusion that the infection of the pharynx is the cause of the difference in lifespan between N2H and N2M.

Major Comments:

This paper is pretty close to rock solid, but there are some concerns that seem like they can be experimentally eliminated.

Antibiotics may have pleiotropic effects –e.g, Ryu et al 2017, Scientific Reports. They induce gene expression changes. The effects of antibiotics ameliorating lifespan differences may be due to genetic background x antibiotic effects in worms or bacteria, and not the infection of the pharynx. The differences in the infection of the pharynx may only correlate with the difference in lifespan. The identical lifespans of N2H and N2M on antibiotics may be caused by different reactions to the antibiotic, resulting in the same lifespan; the ameliorated difference in pharyngeal infection between N2H and N2M may just correlate with the ameliorated differences in lifespan. We cannot be certain with the evidence we have. Furthermore, in addition to worm genetic background differences, the *E. coli* may be different between labs. Would that not be awful? What if we find out sirtuin overexpression in worms is bacterial strain dependent. There are ways to address this: Does culture on the new liposome based axenic media ameliorate pathological and lifespan differences between N2M and N2H? Do animals grown on UV-killed bacteria have no differences in lifespan? How robust is this perhaps important effect?

The authors use old multicopy rol-6 containing sir-2.1 overexpressing strains. The rationale is clear; they are the same old strains used in prior reports. Yet they may be inserted anywhere in the genome, perhaps disrupting other genes. And they have the same suboptimal transgene configuration that causes animals to perpetually twist. It is a very strong, and distinct from wild-type, behavioral phenotype, which must also be consequential to the physiology of the animals. My concern is that, in another decade or so, I do not want to also find out that sirtuin over expression only works to extend lifespan if the worms are rolling. It is 2019. Single copy reporter gene technology with control of locus and copy number has been around since 2008 (MosSCI), and CRISPR based genome editing has been working in worms since 2013. For this to be in Nature Communications, it seems worth verifying with a new, better transgene construct. Does sirtuin overexpression still extend lifespan when worms are not perpetually rolling? Please do not just RNAi rol-6. Just make a clean new strain. If increasing SIR-2.1 expression level has a robust effect on lifespan, it should work. If it does not, that is important too.

Minor Comments:

Figure 1d. Why did red+dark red portion decrease in N2M between day 8 and 12? Is it longitudinal observation or cross-sectional examination of different cohorts? It seemed to me first that this is the result of a longitudinal experiment, and therefore the fraction of red+dark red portion must increase with time. Please provide a better description in materials and methods and figure legend.

Figure 1f. While I somewhat understand the figure, it is a bit confusing. I assume that authors established backcrossed lines by randomly picking a single F2 animal from backcross and used these lines for experiments and successive backcrossings, but I am not 100% certain. Please provide better description in material and methods and figure legends.

The FUDR references for *C. elegans* are incomplete. FUDR is a thymidylate synthetase inhibitor that affects DNA, but has recently been shown to also affect the abundance of rRNA (Burnaevskiy et al), and mitochondrial mutant (*gas-1*) lifespan (Van Raamsdonk and Hekimi).

Given that this paper is on variability in lifespan effects and the importance of genetic background, it might also be worth citing the 2017 Driscoll/Lithgow/Phillips Nat Comm paper on the exact same subject.

In the last paragraph of the results section, the authors erroneously refer to Figure 4 instead of Figure 5.

Reviewer #2 (Remarks to the Author):

This is a very comprehensive mapping of a genetic variant in so called wild type *C. elegans* strains that has a dramatic effect on lifespan. The paper convincingly shows that an *fln-2* nonsense allele causes the increased longevity, via a combination of mapping, crispr reconstruction of the allele in a clean background, and testing of other *fln-2* alleles.

The added exploration of how this allele interacts with a complex and controversial *sir-2* literature for worms is very welcome as well. Overall the paper is ready to publish as is. Very high standards in this paper. Congratulations to the authors.

Author responses to reviewer comments

A common *fln-2* mutation affects lethal pathology and lifespan in *C. elegans*

Reviewer #1 (Remarks to the Author):

In their manuscript titled “A common *fln-2* mutation affects lethal pathology and lifespan in *C. elegans*”, Zhao and colleagues uncovered a confounding factor that affects lab-to-lab and paper-to-paper variability in *C. elegans* lifespan. The authors took a genetic and rudimentary pathological approach to understand differences in lifespan and mortality in different genetic backgrounds in *C. elegans*. They found that a previously undocumented mutation (*fln-2*) in a male stock of *C. elegans* (N2M) was responsible for the lifespan differences between two *C. elegans* stocks (males (N2M) and hermaphrodites (N2H)). Perhaps more importantly, the report addresses conflicting reports on the effects of different genes on lifespan, with *sir-2.1* overexpression being the most potent example. Specifically, the authors addressed an old controversy on the effect of *sir-2.1* overexpression on *C. elegans* lifespan. I would like to thank Dr. David Gems for the honesty and carefulness with which he revisited his Burnett et al, 2011 publication to further clarify the old controversy regarding *sir-2.1*.

The work is well executed and well communicated. It is mostly technically solid; it is supported by a significant body of experimental evidence. There are other technical concerns regarding the conclusions drawn from antibiotics and a roller background (see major comments below). Overall, it is a nice example of a well written, conscientious study (mostly). The results are not exciting or unexpected, but do address previous disparities between reports caused by flawed experimental design. The prior report’s flaw stemmed from the Gems lab practice of outcrossing their animals with the N2M stock instead of just generating males from the N2H, like most *C. elegans* labs of which I am aware. Thus, they compared animals crossed with genetically distinct stocks, with clearly different phenotypes (in retrospect), which ended up having different mutations, causing them to dispute the effects of sirtuin overexpression. Given that the stocks were supposed to be isogenic, this seems forgivable, but also seems like it could have been avoided by more cautious experimentation (deriving males from the strains they have). The importance of genetic background has been abundantly clear for years in *Drosophila*, and to all geneticists. Again, in my experience, most people do not use the *C. elegans* male stock, they just generate males from their existing N2H stock. So, the broad relevance of differences in these stocks, even to gerontologists, is questionable. Yet, Nat Comm is for specialists and the relevance of clarifying conflicting Nature reports on the effects sirtuin overexpression in this system, especially from the investigator who disputed it before, may bring the relevance of the manuscript to a higher level. I think it might be more suitable for Scientific Reports, because we are not really learning any new exciting biology, but my peers and the editor might disagree. And I am a bit on the fence about the publication at Nat Comm, especially given the 2017 publication on the same subject in worms (Lucanic et al); yet, it might be important to show this as an example of science self-correcting. Regardless of where it ends up, there are some technical concerns to address about the biological conclusion that the infection of the pharynx is the cause of the difference in lifespan between N2H and N2M.

Authors response: We thank the reviewer for their kind remarks about this study. Regarding impact: it is true that this study is about confounding effects of background variation but, arguably, a number of other aspects add to its interest. Firstly, we show that *fln-2(ot611)* is present in many strains provided by the *Caenorhabditis* Genetics Center; in fact, of the 90 strains we sequenced so far (not constructed or backcrossed by the Gems lab), over 30% (34 strains) carry the *fln-2(ot611)* allele, presumably due to the use of N2M in other labs. Secondly, the *fln-2* variation affects not just lifespan but also aspects of immunity and behaviour, which make it potentially of interest for non-biogerontological *C. elegans* researchers. Last but not least, the study illustrates the power of new, pathology-focused approaches (here mortality deconvolution) to solve problems in biogerontology, not only to discover the *fln-2* mutation

(which is interesting in its own right), but also to reveal how *sir-2.1* over-expression acts synergistically with FUDR to increase lifespan by reducing P death.

Major Comments:

This paper is pretty close to rock solid, but there are some concerns that seem like they can be experimentally eliminated.

Antibiotics may have pleiotropic effects –e.g, Ryu et al 2017, Scientific Reports. They induce gene expression changes. The effects of antibiotics ameliorating lifespan differences may be due to genetic background x antibiotic effects in worms or bacteria, and not the infection of the pharynx. The differences in the infection of the pharynx may only correlate with the difference in lifespan. The identical lifespans of N2H and N2M on antibiotics may be caused by different reactions to the antibiotic, resulting in the same lifespan; the ameliorated difference in pharyngeal infection between N2H and N2M may just correlate with the ameliorated differences in lifespan. We cannot be certain with the evidence we have. Furthermore, in addition to worm genetic background differences, the *E. coli* may be different between labs. Would that not be awful? What if we find out sirtuin overexpression in worms is bacterial strain dependent. There are ways to address this: Does culture on the new liposome based axenic media ameliorate pathological and lifespan differences between N2M and N2H? Do animals grown on UV-killed bacteria have no differences in lifespan? How robust is this perhaps important effect?

Authors response: To exclude the possibility that suppression of the N2M, N2H lifespan difference by antibiotics was due to effects other than suppression of pharyngeal infection, we compared their lifespans when cultured on UV irradiated *E. coli*, and it was not different. We have added reference to this as follows. "This predicts that blocking *E. coli* proliferation with antibiotics or UV radiation, which completely prevents P death⁹, should abrogate the difference in lifespan between N2H and N2M, and this proved to be the case"

Given that heavy bacterial infection immediately precedes death in P worms, and this is prevented by antibiotic or UV treatment (Zhao *et al.* 2017), and given that the difference in N2M and N2H lifespan is attributable solely to a difference in P frequency and that there is no difference in p lifespans, the deduction that antibiotics and UV irradiation remove the difference in N2M and N2H lifespan by preventing P is not one that involves a major burden of proof. Hence we believe that it would be superfluous to perform the suggested test with culture on liposome-delivered axenic medium.

The authors use old multicopy *rol-6* containing *sir-2.1* overexpressing strains. The rationale is clear; they are the same old strains used in prior reports. Yet they may be inserted anywhere in the genome, perhaps disrupting other genes. And they have the same suboptimal transgene configuration that causes animals to perpetually twist. It is a very strong, and distinct from wild-type, behavioral phenotype, which must also be consequential to the physiology of the animals. My concern is that, in another decade or so, I do not want to also find out that sirtuin over expression only works to extend lifespan if the worms are rolling. It is 2019. Single copy reporter gene technology with control of locus and copy number has been around since 2008 (MosSCI), and CRISPR based genome editing has been working in worms since 2013. For this to be in Nature Communications, it seems worth verifying with a new, better transgene construct. Does sirtuin overexpression still extend lifespan when worms are not perpetually rolling? Please do not just RNAi *rol-6*. Just make a clean new strain. If increasing SIR-2.1 expression level has a robust effect on lifespan, it should work. If it does not, that is important too.

Authors response: In our study, the main claim relating to *sir-2.1* over-expression is that the discrepancy between the previous studies regarding the effect of the transgene array *geIs3* was attributable to variation at *fln-2* (and not *rol-6*). Here the reviewer is considering a different question: under what conditions does *sir-2.1* over-expression increase lifespan? In our submitted manuscript we already demonstrate that effects of *geIs3* on lifespan is condition dependent: it only increases lifespan when FUDR is present. The question of the utility of studying such condition-dependent effects is addressed in the discussion (passage beginning "Does this FUDR-dependence invalidate...").

However, in response to the issue raised, as a further test of condition dependence of the effects of *sir-2.1* over-expression on lifespan, we have now conducted additional tests on non-Rol, low copy number *sir-2.1* over-expression lines created using biolistic transformation (Viswanathan *et al.*, *Dev Cell*. 2005 9, 605-615). We previously reported that backcrossing abrogated life extension in this strain (Burnett *et al.*, *Nature* 2011, 477, 482-485). We have added reference to this new work in the paper as follows.

"We also tested for interaction between FUDR and *sir-2.1* over-expression using a previously described low copy number transgene array *pkIs1642*²⁰. It was previously reported that longevity resulting from *pkIs1642 sir-2.1(oe)* was abrogated by outcrossing²¹. Here both original and outcrossed strains were *fln-2(+)*. Nonetheless, we observed that the longevity of the non-outcrossed *pkIs1642* strain was, again, FUDR dependent (Supplementary Fig. 6; Supplementary Table 8). However, here longevity resulted from increased p lifespan rather than reduced P frequency. For more details, see Supplementary Fig. 6."

Minor Comments:

Figure 1d. Why did red+dark red portion decrease in N2M between day 8 and 12? Is it longitudinal observation or cross-sectional examination of different cohorts? It seemed to me first that this is the result of a longitudinal experiment, and therefore the fraction of red+dark red portion must increase with time. Please provide a better description in materials and methods and figure legend.

Authors response: Data shown in Fig. 1d is a cross-sectional examination of different cohorts, as these worms had to be carefully examined under high magnification fluorescent microscope for signs of infection, and did not recover well after imaging. A better description of the experimental procedure is now provided in the Material and Methods, as follows. "RFP infection analysis. L4 worms cultured on OP50 bacteria were transferred to OP50-RFP on NGM, and transferred daily during the reproductive period. Before imaging, live worms were mounted on 2% agar pads and anesthetised in 0.2% levamisole. Fluorescence within pharyngeal tissue were scored at 630x magnification using a Zeiss AxioImager Z2 microscope with a rhodamine filter. A different cohort of worms was imaged for each time point."

Figure 1f. While I somewhat understand the figure, it is a bit confusing. I assume that authors established backcrossed lines by randomly picking a single F2 animal from backcross and used these lines for experiments and successive backcrossings, but I am not 100% certain. Please provide better description in material and methods and figure legends.

Authors response: A better description of the experimental procedure is now provided in the Materials and Methods section, as follows. "Test cross to confirm that N2M reduced P phenotype is associated with a single locus. Strain GR1307 *daf-16(mgDf50)* was backcrossed with either N2H or N2M males, and F1s allowed to self. Single F2 animals were picked at random and the *daf-16* allele genotyped, and F3 populations of single *daf-16* homozygous animals were scored in necropsy analysis and used for further backcrossing. The same analyses were performed for a single F4, F6, and F8 animal of each F2 line. All progeny of the backcrossed F2, F4, F6, and F8 animals used in lifespans were frozen and subsequently genotyped with respect to the *fln-2* allele."

The FUDR references for *C. elegans* are incomplete. FUDR is a thymidylate synthetase inhibitor that affects DNA, but has recently been shown to also affect the abundance of rRNA (Burnaevskiy *et al*), and mitochondrial mutant (*gas-1*) lifespan (Van Raamsdonk and Hekimi).

Authors response: We thank the reviewer for pointing out these studies. They are now cited.

Given that this paper is on variability in lifespan effects and the importance of genetic background, it might also be worth citing the 2017 Driscoll/Lithgow/Phillips *Nat Comm* paper on the exact same subject.

Authors response: Good idea. We have now cited this paper.

In the last paragraph of the results section, the authors erroneously refer to Figure 4 instead of Figure 5.

Authors response: Fixed.

Reviewer #2 (Remarks to the Author):

This is a very comprehensive mapping of a genetic variant in so called wild type *C. elegans* strains that has a dramatic effect on lifespan. The paper convincingly shows that an *fln-2* nonsense allele causes the increased longevity, via a combination of mapping, crispr reconstruction of the allele in a clean background, and testing of other *fln-2* alleles.

The added exploration of how this allele interacts with a complex and controversial *sir-2* literature for worms is very welcome as well. Overall the paper is ready to publish as is. Very high standards in this paper. Congratulations to the authors.

Author response: We thank the reviewer for their very positive assessment of this study.

REVIEWERS' COMMENTS:

Reviewer #1 (Remarks to the Author):

Good job and congratulations to the authors. The authors have done a sufficient job addressing technical concerns. The scientific findings are now improved via being better supported by additional experimental evidence. The additional UV-killed bacteria lifespans and additional old sir2.1 overexpression strains add said support.

This work is up to par with the scientific standards of the *C. elegans* aging field.

I do wish that the authors had performed the overexpression experiments in a more technically solid fashion; but everyone cannot do everything. Specifically, they used old existing reporters instead of making new ones, and found the same FUDR-dependent lifespan effect in a non-roller background. It would be nice to just see what adding extra copies of the sir-2.1 gene does in a coherent, one at a time fashion. We only know about funky multi-copy gene insertions as it stands, albeit some are lower copy number.

The paper makes important points about poor and common technical practices in *C. elegans* aging research.

Overall, the authors have done a good job addressing scientific concerns and the paper deserves to be published. I still have the same reservations about this paper being mostly technical biology about a relatively small lifespan effect. However, the paper is scientifically better supported than many worm lifespan papers. And, since I am reviewing the paper again, I presume the editor also finds this paper scientifically righteous and/or interesting. It is a good example of science correcting itself.

Author responses to reviewer comments

A fln-2 mutation affects lethal pathology and lifespan in *C. elegans*

REVIEWERS' COMMENTS:

Reviewer #1 (Remarks to the Author):

Good job and congratulations to the authors. The authors have done a sufficient job addressing technical concerns. The scientific findings are now improved via being better supported by additional experimental evidence. The additional UV-killed bacteria lifespans and additional old sirtuin overexpression strains add said support.

This work is up to par with the scientific standards of the *C. elegans* aging field.

I do wish that the authors had performed the overexpression experiments in a more technically solid fashion; but everyone cannot do everything. Specifically, they used old existing reporters instead of making new ones, and found the same FUDR-dependent lifespan effect in a non-roller background. It would be nice to just see what adding extra copies of the *sir-2.1* gene does in a coherent, one at a time fashion. We only know about funky multi-copy gene insertions as it stands, albeit some are lower copy number.

The paper makes important points about poor and common technical practices in *C. elegans* aging research.

Overall, the authors have done a good job addressing scientific concerns and the paper deserves to be published. I still have the same reservations about this paper being mostly technical biology about a relatively small lifespan effect. However, the paper is scientifically better supported than many worm lifespan papers. And, since I am reviewing the paper again, I presume the editor also finds this paper scientifically righteous and/or interesting. It is a good example of science correcting itself.

Authors response: We thank the reviewer for their previous comments which helped make this paper more scientifically rigorous. We also thank the reviewer for their kind remarks about the revised manuscript.